# The Fact of Return to Work in Cervical Cancer Survivors and the Impact of Survival Rate: An 11-Year Follow-Up Study

**DOI:** 10.3390/ijerph182010703

**Published:** 2021-10-12

**Authors:** Yu-Shan Sun, Wei-Liang Chen, Wei-Te Wu, Chung-Ching Wang

**Affiliations:** 1Division of Family Medicine, Department of Family and Community Medicine, Tri-Service General Hospital, National Defense Medical Center, Taipei 114, Taiwan; DOC10685@mail.ndmctsgh.edu.tw (Y.-S.S.); weiliang0508@gmail.com (W.-L.C.); 2Division of Environmental Health & Occupational Medicine, Department of Family and Community Medicine, Tri-Service General Hospital, National Defense Medical Center, Taipei 114, Taiwan; 3Graduate Institute of Medical Sciences, National Defense Medical Center, Taipei 114, Taiwan; 4National Institute of Environmental Health Sciences, National Health Research Institutes, Miaoli 350, Taiwan; ader.una@gmail.com

**Keywords:** return to work, cervical cancer, cancer survivors, survival rate

## Abstract

The aim of the current cohort study was to explore the relationship between return to work (RTW) after cervical cancer treatment and different medical and occupational covariates. We also investigated the effect of RTW on all-cause mortality and survival outcomes of cervical cancer survivors. Data were collected between 2004 and 2015 from the database of the Taiwan Cancer Registry, Labor Insurance Database, and National Health Insurance Research Database. The associations between independent variables and RTW were analyzed by Cox proportional hazard models. A total of 4945 workers (82.3%) who returned to work within 5 years after being diagnosed with cervical cancer. Patients who underwent surgical treatment were more likely to RTW by the 5th year compared to other groups, with a hazard ratio (HR) of 1.21 (95% CI: 1.01~1.44). Small company size and a monthly income greater than NT 38,200 were inversely associated with RTW (HR = 0.91, 95% CI: 0.84~0.98 and HR = 0.48, 95% CI: 0.44~0.53). Furthermore, RTW showed a statistically significant decrease in the risk of all-cause mortality in the fully adjusted HR, (HR = 0.42, *p* < 0.001). Some medical and occupational factors are associated with RTW in cervical cancer survivors. Returning to work may have a beneficial effect on the survival of patients with cervical cancer.

## 1. Introduction

In most countries, cancer is viewed as the most prevalent cause of morbidity and mortality. The treatment of cancer not only affects the physical condition of the patients but also affects their mental status, occupational status and quality of life [1,2,3]. Most patients take sick leave after the first cancer diagnosis [4]. With the improvement of cancer treatment, patients have a better prognosis after treatment and tend to return to work (RTW). RTW after cancer treatment is an important symbol of disease control and the patient getting back to a semblance of normal life. Several studies report that multiple factors, including work environment, personal condition, and disease condition, were associated with cancer survivors’ willingness to RTW [5,6]. Multidisciplinary interventions that involve physical, psychoeducational, and occupational aspects may enhance cancer survivors’ RTW [7]. However, the overall outcome of cancer survivors after returning to work remains unclear.

Cervical cancer is one of the most frequent cancers in women of working age. Cervical cancer treatment-related adverse effects, such as rectal bleeding and urinary self-catheterization, may influence the employment condition of the patients [8].With comprehensive cancer control strategies, the survival rate of cervical cancer has increased in recent years, particularly in well-developed countries [9]. More women tend to RTW after treatment of cervical cancer. The factors that are associated with RTW condition may differ by gender. However, most previous studies on women returning to work after cancer treatment have focused on breast cancer survivors [10,11,12]. The aim of the current cohort study was to explore the relationship between RTW and covariables, including sociodemographic characteristics, cancer treatment, and occupation factors, in cervical cancer survivors. We also investigated the effect of RTW on all-cause mortality and survival outcomes of workers with cervical cancer.

## 2. Materials and Methods

### 2.1. Study Population

For this study, data were collected from participants who were diagnosed with cervical cancer between 2004 and 2015 according to the Taiwan Cancer Registry (TCR) database. TCR is a nationwide population-based cancer registry system that collected the data of patients with newly diagnosed cancer in hospitals with 50 or more beds in Taiwan. We then linked the participant data to the Labor Insurance Database (LID) registry and the National Health Insurance Research Database (NHIRD) by a unique encryption identity number. The study was approved by the Institutional Review Board of Tri-Service General Hospital (IRB No 1-107-05-129). Participants who were younger than 20 years old, had cervical cancer combined with other cancers, were unemployed, or had missing data in any of the databases mentioned above were excluded from our study.

### 2.2. Diagnosis of Cervical Cancer

The diagnostic and staging criteria of cervical cancer were according to the American Joint Committee on Cancer (AJCC) TNM classification 7th edition [13]. We included workers who were newly diagnosed with cervical cancer based on the ICD-O (including ICD-O-3 C53.0, C53.1, C53.8, and C53.9) recorded in the TCR database during the study period of 2004–2015. Primary treatments for cervical cancer, such as surgical intervention, chemotherapy, or radiation therapy, were obtained from the NHIRD database.

### 2.3. Assessment of Covariates

Demographic data, such as age, employment area, company size, and monthly income were collected from the LID. Data on other clinical comorbidities, including disorders of lipid metabolism, obesity, hypertension, cerebrovascular disease, congestive heart failure, peptic ulcer disease, chronic pulmonary disease, liver disease, renal disease, and depression, were collected from the NHIRD and reported with International Classification of Diseases, Ninth Revision, Clinical Modification (ICD-9-CM) codes. The ICD-9-CM codes were as follows: lipid metabolism (ICD-9-CM code 272), obesity (ICD-9-CM code 278), hypertension (ICD-9-CM codes 401.x–405.x), cerebrovascular diseases (ICD-9-CM codes 362.34, 430.x–438.x), chronic pulmonary diseases (ICD-9-CM codes 416.8, 416.9, 490.x–505.x, 506.4, 508.1, 508.8), peptic ulcer diseases (ICD-9-CM codes 531.x–534.x), renal diseases (ICD-9-CM codes 403.01, 403.11, 403.91, 404.02, 404.03, 404.12, 404.13, 404.92, 404.93, 582.x, 583.0–583.7, 585.x, 586.x, V42.0, V45.1, V56.x), liver diseases (ICD-9-CM codes 070.22, 070.23, 070.32, 070.33, 070.44, 070.54, 070.6, 070.9, 570.x, 571.x, 573.3, 573.4, 573.8, 573.9, V42.7), psychoses (ICD-9-CM codes 293.8, 295.x, 296.04, 296.14, 296.44, 296.54, 297.x, 298.x), and depression (ICD-9-CM codes 296.2, 296.3, 296.5, 300.4, 309.x, 311.x).

### 2.4. Outcome Measures

The primary outcome for this study was RTW within the first five years after cervical cancer diagnosis. RTW was defined according to employment status in LID records. The secondary outcome was all-cause mortality within the follow-up period after RTW. The mortality data were obtained from NHIRD.

### 2.5. Statistical Analysis

The SAS statistical software package (version 9.3, SAS Institute Inc., Cary, NC, USA) was used for analyses in our study. Descriptive statistical analyses, such as the mean, percentage, and standard deviation, were determined for all the identified variables. The chi-squared test for categorical variables and Wilcoxon rank sum test or independent *t*-test for continuous variables were applied to analyze the distribution of covariates across subgroups. Two-sided *P* values of less than 0.05 were considered to indicate significance. RTW was counted from the date of cervical cancer diagnosis to the date of reemployment within five years. Survival time was computed from the date of cervical cancer diagnosis to the date of death during the period of 2004–2015. Kaplan-Meier survival curves were plotted to ascertain the relationship of the RTW and subsequent mortality categorized by different stages of cervical cancer. Cox proportional hazard regression was used to determine the effects of different variables on the RTW and survival outcomes.

## 3. Results

### 3.1. Demographic Characteristics of Cervical Cancer Workers with and without RTW

The cervical cancer workers’ demographic characteristics, such as age, pathological stage of the tumor, clinical comorbidities, living area, monthly income, industrial classification, and company size, are listed in Table 1. Among the 6008 participants, 4945 workers (82.3%) returned to work within 5 years after cervical cancer diagnosis. The mean ages of the RTW group and non-RTW group were 43.6 and 47.1 years, respectively. In the RTW group, most of the cancer survivors were in the pathological stage 0 (82.5%), followed by stage 1 (14.3%). Most cancer survivors in the RTW group underwent surgical intervention (97.5%).

### 3.2. Characteristics of Employment of Cervical Cancer Survivors during the 5-Year Period

Table 2 presents the clinical features and several work factors of RTW workers with cervical cancer. There were 5049 workers (84.0%) continuing ordinary work or returning to work after treatment in the first 2 years. A total of 75.3% of the RTW workers’ monthly income was less than NT 28,800. The most common industrial classification of RTW workers was manufacturing (31.1%), followed by wholesale and retail trade (14.7%). More than half of the RTW workers were employed at a large company.

### 3.3. Univariate Association between RTW and Independent Variables

The unadjusted models of hazard ratios (HRs) for RTW in different years are listed in Appendix A Table A1. Demographic characteristics, including age and therapeutic treatments with radiation and chemotherapy, were negatively associated with RTW (*p* < 0.05). Participants who underwent surgical treatment were more likely to RTW, with an HR of 1.19 (95% CI: 1.00~1.41) in the 2nd year. The HR was slightly elevated by year, with an HR of 1.32 (95% CI: 1.11~1.58) in the 5th year. Compared with stage 3 and 4 disease, employees with stage 0 and stage 1 disease were significantly more likely to RTW with an HR of 1.40 (95% CI: 1.18~1.65) and 1.38 (95% CI: 1.16~1.65), respectively, in the 2nd year, then with an increased HR of 2.25 (95% CI: 1.84~2.76) and 2.00 (95% CI: 1.61~2.47), respectively, in the 5th year.

### 3.4. Multivariate Association between RTW and Independent Variables

The multivariate Cox proportional hazard model showed that patients who underwent surgical treatment were more likely to RTW by the 5th year compared to other groups, with an HR of 1.21 (95% CI: 1.01~1.44) (Appendix A Table A2). Compared with patients with stage 3 and 4 disease, patients with stage 0 and stage 1 disease showed a significantly increased likelihood to RTW, with HRs of 1.97 (95% CI: 1.52~2.54) and 1.80 (95% CI: 1.41~2.31), respectively, even after completely adjusting for multiple covariates. Compared with large companies, small to medium company sizes were inversely associated with RTW, with an HR of 0.91 (95% CI: 0.84~0.98). Workers with a monthly income greater than NT 38,200 were less likely to RTW, with an HR of 0.48 (95% CI: 0.43~0.53). However, the results showed no significant association with industrial classification and RTW.

### 3.5. Effect of RTW on Survival Outcome in Cervical Cancer Survivors

The survival rates of all cervical cancer survivors were significantly higher in the RTW group than in the non-RTW group (*p* < 0.001) (Figure 1). Kaplan-Meier survival analysis demonstrated that the RTW group had significantly better survival than the non-RTW group in patients with stage 0 (Figure 2A) (*p* = 0.039) and stage 3 and 4 (Figure 2D) (*p* < 0.001) cervical cancer. For stage 1 (Figure 2B) and stage 2 (Figure 2C) patients, there was no significant difference between the RTW group and the non-RTW group. After Cox proportional hazard analysis (Table 3), the fully adjusted HR indicated that the risk of all-cause mortality was significantly reduced when the patient returned to work (HR = 0.42, *p* < 0.001).

## 4. Discussion

This cohort study investigated the multifaceted factors that may be related to the RTW status of cervical cancer patients and the survival outcome in those patients within the 5-year follow-up period. We found that younger patients and patients who only underwent surgical treatment had more willingness to RTW. Regarding employment factors, workers who worked in small companies or had higher incomes were less likely to RTW. Regarding survival outcomes, the survival rates of the RTW group were significantly higher than those of the non-RTW group. Furthermore, according to Cox proportional hazard analysis, the presence of RTW showed a statistically significant decrease in the risk of all-cause mortality in the fully adjusted HR.

With the increased cancer survival rate, employment status after cancer treatment has been demonstrated in several studies. A systematic review of sixty-four studies showed that an average of 89% of employees had returned to work within 2 years after their cancer diagnosis [14]. A Japanese cohort study reported that different cancer types were significantly different in RTW rates. Patients with gastric cancer, intestinal cancer, breast cancer, male/female genital cancer, or urinary cancer had higher RTW rates than the patients with lung cancer, esophageal cancer, hepatic cancer, pancreatic cancer, or blood cancer [15]. In our study, 84% of participants returned to work in the first two years of the follow-up period, and patients who underwent only surgical treatment were more likely to RTW. A study of cervical cancer survivors in Japan showed that patients who had undergone radical hysterectomy plus concurrent chemoradiation/radiation therapy were less likely to RTW after treatment due to the complication of lower extremity edema [16]. Employment factors, such as working conditions, work demands and flexibility, may be associated with RTW [5]. The current study showed that workers who worked in small companies were less likely to RTW. A study including 3024 employees in the Netherlands reported that cancer survivors working in small companies returned to work later than those working in large companies [17]. Generally, compared with small employers, large employers may be more able to accommodate employees. Several previous breast cancer survivor studies demonstrated that workplace policies, such as rehabilitative interventions or support from employers and coworkers, enhance RTW [11,18].

Income has been found to be a predictive factor of employment status in cancer survivors [19]. A higher income level may be associated with a higher education level and lower physical job demands. A large population cohort study in Europe reported that a lower income was one of the risk factors for unemployment of cancer survivors [20]. However, a study of low-income Latina breast cancer survivors demonstrated that in the first 18 months after diagnosis, fewer low-income Latinas returned to work than non-Latina Caucasian patients, but the RTW rate was not significantly different at 36 months after diagnosis [21]. A Korean retrospective cohort study found that a lower income level was a predictor of early job loss but had no relation to the time required to RTW [22]. Our study reported that those who had a higher income were less likely to RTW at both one year and five years after treatment (Appendix A Figure A1). According to the Manpower Survey Results of Taiwan in 2014, the labor force participation rate of women aged 45–54 years was 63.8%, which is lower than in the United States (74.0%), Canada (80.1%), and the United Kingdom (82.1%). The results may indicate that women aged 45–54 in Taiwan participate less in work than Western countries and seem to be more economically dependent on their family [23,24]. Thus, the higher income group might represent a higher cumulative income that led to a longer leave from work. On the other hand, the lower income group might have needed to RTW earlier due to the family financial burden.

With screening programs for the early detection of cancer and improvements in cancer treatment, the number of working-age cancer survivors has increased. Several studies have addressed the impact of returning to work on cancer survivors, such as changes in working ability [25], income [26], and quality of life [27,28,29]. However, our study is the first to analyze the survival outcome in cervical cancer survivors after returning to work. In fully adjusted Cox proportional hazard analysis, the presence of RTW was significantly decreased in the risk of all-cause mortality. This finding may indicate that RTW in cancer survivors not only represents returning to normal life but can also improve the long-term survival rate. Lindsay et al. reported that compared with gynecologic cancer survivors who RTW, those who were not employed were at nearly 3.5 times greater risk of high distress [30]. A German breast cancer survivor study revealed that women who permanently stopped working showed worse quality of life and cognitive status over time than women who worked as they had worked before diagnosis [27]. Mahar et al. reported that women who discontinued employment after a cancer diagnosis had higher psychosocial distress level and worse physical and mental functioning and quality of life than women who continued working during treatment and those who stopped working during treatment but returned to work [31].

There are some limitations to this study. First, some demographic information, such as educational level and marital status, which may be associated with RTW, was not available in the databases. Second, the psychological condition of the participants was not included in the databases, and the psychological impact of RTW and overall survival were not investigated in the current study. Third, the period of cancer treatment was not included in the database, which may be related to RTW and the overall survival. Fourth, the LID and NHIRD databases used in our study were based on a national cancer screening program; however, the screening program was created for epidemiological research, and the inference of occupational exposure and environmental impact cannot be clearly identified.

## 5. Conclusions

In the current study, we investigated multiple covariates, including disease condition, sociodemographic status and occupational factors that may be associated with the RTW status of cervical cancer survivors. We also demonstrated that RTW may have a positive effect on cancer survivors in terms of their overall survival outcome. Interventions to improve the cancer survivors’ ability to RTW should be emphasized after cancer treatment. Additional studies of prognostic conditions, such as physical and psychological impacts after RTW, should be considered.

## Figures and Tables

**Figure 1 ijerph-18-10703-f001:**
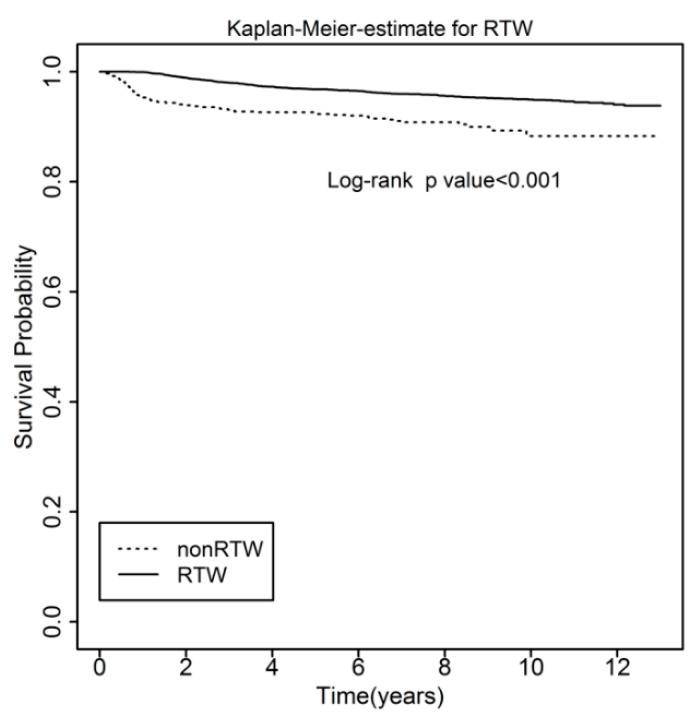
Kaplan-Meier curve for all-cause mortality in the workers with all stages of cervical cancer.

**Figure 2 ijerph-18-10703-f002:**
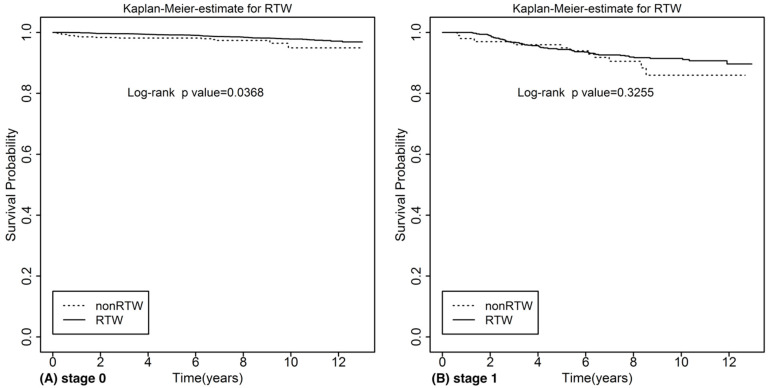
Kaplan-Meier curve for all-cause mortality categorized by different stages of cervical cancer. (**A**) Stage 0 of cervical cancer (**B**) Stage 1 of cervical cancer (**C**) Stage 2 of cervical cancer (**D**) Stage 3&4 of cervical cancer.

**Table 1 ijerph-18-10703-t001:** Demographic data of RTW group and non-RTW group of cervical cancer workers.

Characteristic	RTW (*n* = 4945)	%	Non-RTW (*n* = 1063)	%	*p* Value
Age (years)	43.6 ± 9.0		47.1 ± 10.1		<0.001
Comorbidities					
Disorders of lipid metabolism	202	4.1%	66	6.2%	0.002
Obesity	13	0.3%	4	0.4%	0.524
Hypertension	397	8.0%	134	12.6%	<0.001
Congestive heart failure	16	0.3%	10	0.9%	0.016
Cerebrovascular disease	34	0.7%	13	1.2%	0.072
Chronic pulmonary disease	110	2.2%	37	3.5%	0.016
Peptic ulcer disease	232	4.7%	64	6.0%	0.069
Liver disease	177	3.6%	50	4.7%	0.081
Renal disease	31	0.6%	21	2.0%	<0.001
Depression	116	2.4%	33	3.1%	0.149
Treatment					<0.001
Operation	4820	97.5%	1002	94.26%	
Radiation therapy	228	4.6%	182	17.12%	
Chemotherapy	128	2.6%	111	10.44%	
Living area when diagnosed with cancer					0.265
Central	1262	25.5%	243	22.9%	
North	2448	49.5%	563	53.0%	
East	83	1.7%	20	1.9%	
South+ Islands	1152	23.3%	237	22.3%	
Monthly income (TWD)					<0.001
<28,800	3722	75.3%	543	51.1%	
28,800–38,200	696	14.1%	124	11.7%	
>38,200	527	10.7%	396	37.2%	
Industrial classification					0.516
Agriculture, Forestry, Fishing, and Animal Husbandry	347	7.0%	63	5.9%	
Manufacturing	1539	31.1%	311	29.3%	
Construction	403	8.2%	88	8.3%	
Wholesale and Retail Trade	728	14.7%	152	14.3%	
Transportation and Storage	165	3.3%	39	3.7%	
Accommodation and Food Service Activities	272	5.5%	55	5.2%	
Information and Communication	82	1.7%	23	2.2%	
Financial and Insurance Activities	192	3.8%	56	5.3%	
Real Estate Activities	63	1.3%	12	1.1%	
Professional, Scientific, and Technical Activities	104	2.1%	36	3.4%	
Support Service Activities	114	2.3%	25	2.4%	
Public Administration and Defense	68	1.4%	15	1.4%	
Education	102	2.1%	19	1.8%	
Human Health and Social Work Activities	152	3.1%	39	3.7%	
Amusement and Recreation Activities	74	1.5%	16	1.5%	
Other Service Activities	515	10.4%	109	10.3%	
Company size					0.124
Missing	452	9.1%	94	8.8%	
Small	369	7.5%	85	8.0%	
Small and medium	1196	24.2%	291	27.4%	
Large	2928	59.2%	593	55.8%	
pStage					<0.001
0	4079	82.5%	705	66.3%	
1	707	14.3%	197	18.5%	
2	62	1.3%	38	3.6%	
3	89	1.8%	84	8.0%	
4	8	0.2%	39	3.7%	

**Table 2 ijerph-18-10703-t002:** Demographic data of RTW individuals from 2nd and 5th year of cervical cancer workers.

Characteristic	2-Year RTW (*n* = 5049)	%	5-Year RTW (*n* = 4945)	%
Age	44.6 ± 9.1		44.2 ± 8.9	
Comorbidities				
Disorders of lipid metabolism	212	4.2%	202	4.1%
Obesity	13	0.3%	13	0.3%
Hypertension	428	8.4%	397	8.0%
Congestive heart failure	18	0.4%	16	0.3%
Cerebrovascular disease	39	0.8%	34	0.7%
Chronic pulmonary disease	120	2.4%	110	2.2%
Peptic ulcer disease	247	4.9%	232	4.7%
Renal disease	188	3.7%	177	3.6%
Liver disease	32	0.6%	31	0.6%
Depression	122	2.4%	116	2.4%
Treatment				
Operation	4955	97.3%	4820	97.47%
Radiation therapy	305	6.0%	228	4.6%
Chemotherapy	172	3.4%	128	2.6%
Living area when diagnosed with cancer				
Central	1304	25.6%	1262	25.5%
North	2503	49.1%	2448	49.5%
East	85	1.7%	83	1.7%
South + Islands	1202	23.6%	1152	23.3%
Monthly income (TWD)				
<28,800	3809	74.8%	3722	75.3%
28,800–38,200	708	13.9%	696	14.1%
>38,200	577	11.3%	527	10.7%
Industrial classification				
Agriculture, Forestry, Fishing, and Animal Husbandry	364	7.2%	347	7.0%
Manufacturing	1617	31.7%	1539	31.1%
Construction	420	8.2%	403	8.2%
Wholesale and Retail Trade	735	14.4%	728	14.7%
Transportation and Storage	171	3.4%	165	3.3%
Accommodation and Food Service Activities	281	5.5%	272	5.5%
Information and Communication	84	1.7%	82	1.7%
Financial and Insurance Activities	198	3.9%	192	3.9%
Real Estate Activities	63	1.2%	63	1.3%
Professional, Scientific, and Technical Activities	105	2.1%	104	2.1%
Support Service Activities	117	2.3%	114	2.3%
Public Administration and Defense	69	1.4%	68	1.4%
Education	99	1.9%	102	2.1%
Human Health and Social Work Activities	153	3.0%	152	3.1%
Amusement and Recreation Activities	76	1.5%	74	1.5%
Other Service Activities	542	10.6%	515	10.4%
Company size				
Missing	459	9.0%	452	9.1%
Small	375	7.4%	369	7.5%
Small to medium	1207	23.7%	1196	24.2%
Large	3053	59.9%	2928	59.2%
pStage				
0	4107	80.6%	4079	82.5%
1	766	15.0%	707	14.3%
2	76	1.5%	62	1.3%
3	126	2.5%	89	1.8%
4	19	0.4%	8	0.2%

**Table 3 ijerph-18-10703-t003:** Multivariate analysis of association between independent variables and all-cause mortality of cervical cancer workers.

Characteristic	HR (95%CI)	*p* Value
Return to work (ref:non-RTW)	0.42 (0.31~0.58)	<0.001
Age (ref: age <40)		
40 < age <= 48	1.12 (0.79~1.59)	0.522
age > 48	1.65 (1.19~2.28)	0.002
Treatment (ref: no)		
Operation	0.52 (0.34~0.79)	0.002
Radiation therapy	1.21 (1.20~2.42)	0.002
Chemotherapy	1.33 (0.96~1.83)	0.086
Monthly income (ref: <28,800)		
28,800–38,200	1.00 (0.73~1.38)	0.981
>38,200	0.56(0.37~0.86)	0.007
Company size (ref: large)		
Missing	1.12 (0.73~1.72)	
Small	0.69 (0.40~1.17)	
Small and medium	0.93 (0.68~1.26)	
pStage (ref: 3&4)		
0	0.06 (0.04~0.09)	<0.001
1	0.20 (0.14~0.29)	<0.001
2	0.51 (0.33~0.78)	0.002

## Data Availability

Raw data were generated at the National Health Insurance Research Database, Taiwan Cancer Registry Database, and Labor Insurance Database in Taiwan. Derived data that support the findings of this study are available from the corresponding author on request.

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
