# Peer review of "The Fact of Return to Work in Cervical Cancer Survivors and the Impact of Survival Rate: An 11-Year Follow-Up Study"

_ijerph, 2021, doi:10.3390/ijerph182010703_

Round 1

Reviewer 1 Report

Sun et al. studied the relationship between return to work (RTW) after cervical cancer treatment, survival and different medical and occupational covariates. Included were women with a cervical cancer diagnosis between 2004 and 2015, and data were collected from the database of the Taiwan Cancer Registry, Labor Insurance Database, and National Health Insurance Research Database. Of 6,008 patients with cervical cancer, 4,945 women (82.3%) returned to work within 5 years after the diagnosis.

Patients underwent surgical treatment were more likely to RTW by the 5th year compared to women treated by radiation or chemotherapy, with a hazard ratio (HR) of 1.21 (95% CI: 1.01-1.44). RTW showed a statistically significant decrease in the risk of all-cause mortality in the fully adjusted HR, (HR=0.42, p<0.001). In conclusion, returning to work may improve survival of patients with cervical cancer.

The claims are properly placed in the context of the previous literature. The experimental data support the claims. The manuscript is written clearly enough that most of it is understandable to non-specialists. The authors have provided adequate proof for their claims, without overselling them. The authors have treated the previous literature fairly. The paper offers enough details of methodology so that the experiments could be reproduced.

Comments

The manuscript includes too many tables with too many columns, too many numbers and too many digits. This makes the manuscript difficult to read.

  1. I suggest moving table 3 and 4 to the appendix.
  2. In table 3 and 4, delete the results from 1-year, 3-year and 4-years RTW (only keep 2-year RTW and 5-year RTW).
  3. All HR in the text and tables should be presented with two decimals only “1.206 (95% CI: 1.009~1.441)” => “1.21 (95% CI: 1.01-1.44)”
  4. All percentages in the text and tables should be presented with one decimal only, “There were 5,049 workers (84.03%) continuing ordinary work or returning” => “There were 5,049 workers (84.0%) continuing ordinary work or returning”, “A total of 75.27% of the RTW workers’ monthly income was less than NT$28,800” => “A total of 75.3% of the RTW workers’ monthly income was less than NT$28,800”
  5. P-values should mainly be presented with two decimals “0.0023” => “0.002”, “0.5235” => “0.524” (but P < 0.001 is ok).
  6. I am not native English speaking, but I think “explore” is a better term than “discover”, “The aim of the current cohort study was to discover the relationship between” => “The aim of the current cohort study was to explore the relationship between”
  7. Page 11, line 161-162, “For stage 0 (Figure 2A), stage 1 (Figure 2B) and stage 2 (Figure 2C) patients, there was no significant difference between the RTW group and the non-RTW group”. In the Material and Methods section, “Two-sided ? values of less than 0.05 were considered to indicate significance”. For stage 0 the Log-rank p value = 0.0368 (Figure 2A). This is less than 0.05, indicating that women with stage 0 cancer who RTW had significant better survival than stage 0 nonRTW.
  8. Page 14, line 274, “Informed consent was obtained from all subjects involved in the study”, actually I do not believe that you had written consent from all 6,008 women. First of all, the data from the databases had a “encryption identity number”. If you do not know the identity of any women, it is impossible to send invitation letters in the mail. Second, the research received no external founding. It is expensive to send letters to thousands of women. Third, the women got their cancer diagnosis between 2004 and 2015. If the study got its ethical approval in 2018, some women got their cancer diagnosis 14 years earlier, and some of them are dead. Did you get written consents from next of kin in case the women was dead? In my experience, only half of the invited study objects will return written consent in retrospective studies. How many women were invited to participate? Usually there is no need for written consent from subjects when using data from a de-identified database.

Author Response

Thank you for your positive comments on this manuscript. The responses to the points raised by the reviewers are listed as follows.

1. I suggest moving Table 3 and 4 to the appendix.

Response: Thank you for your critique. We followed your recommendation and modified the last paragraph of the appendix section. (Page11-14)

2. In Table 3 and 4, delete the results from 1-year, 3-year, and 4-years RTW (only keep 2-year RTW and 5-year RTW).

Response: Thank you for your valuable comments. We followed your recommendation and modified the last paragraph of the appendix section. (Page11-14)

3. All HR in the text and tables should be presented with two decimals only “1.206 (95% CI: 1.009~1.441)” => “1.21 (95% CI: 1.01-1.44)”

Response: Thank you for your suggestions. We followed your recommendation and modified the last paragraph.

4. All percentages in the text and tables should be presented with one decimal only, “There were 5,049 workers (84.03%) continuing ordinary work or returning” => “There were 5,049 workers (84.0%) continuing ordinary work or returning”, “A total of 75.27% of the RTW workers’ monthly income was less than NT$28,800” => “A total of 75.3% of the RTW workers’ monthly income was less than NT$28,800”

Response: Thank you for your critique. We followed your recommendation and modified the last paragraph.

5. P-values should mainly be presented with two decimals “0.0023” => “0.002”, “0.5235” => “0.524” (but P < 0.001 is ok).

Response: Thank you for your valuable comments. We followed your recommendation and modified the last paragraph.

6. I am not native English speaking, but I think “explore” is a better term than “discover”, “The aim of the current cohort study was to discover the relationship between” => “The aim of the current cohort study was to explore the relationship between”  

Response: Thank you for your thorough review and critique. We followed your recommendation and modified the last paragraph of the abstract and introduction section. (Page 1, line 13 and Page 2, line 68)

7. Page 11, line 161-162, “For stage 0 (Figure 2A), stage 1 (Figure 2B) and stage 2 (Figure 2C) patients, there was no significant difference between the RTW group and the non-RTW group”. In the Material and Methods section, “Two-sided ? values of less than 0.05 were considered to indicate significance”. For stage 0 the Log-rank p value = 0.0368 (Figure 2A). This is less than 0.05, indicating that women with stage 0 cancer who RTW had significant better survival than stage 0 non RTW.

Response: Thank you for your critique. We followed your recommendation and modified the last paragraph of the result section. (Page 7, line 1116-1118)

8. Page 14, line 274, “Informed consent was obtained from all subjects involved in the study”, actually I do not believe that you had written consent from all 6,008 women. First of all, the data from the databases had a “encryption identity number”. If you do not know the identity of any women, it is impossible to send invitation letters in the mail. Second, the research received no external founding. It is expensive to send letters to thousands of women. Third, the women got their cancer diagnosis between 2004 and 2015. If the study got its ethical approval in 2018, some women got their cancer diagnosis 14 years earlier, and some of them are dead. Did you get written consents from next of kin in case the women was dead? In my experience, only half of the invited study objects will return written consent in retrospective studies. How many women were invited to participate? Usually there is no need for written consent from subjects when using data from a de-identified database.

Response: Thank you for your insightful comments and helpful suggestions. Subject informed consent was waived in this study. Because our data were collected from the database of the Taiwan Cancer Registry (TCR), Labor Insurance Database (LID), and National Health Insurance Research Database (NHIRD). Those databases were provided under de-identification status, we can only link the data by a unique encryption identity number. We followed your recommendation and modified the last paragraph of the informed consent statement section. (Page 11, line 1277-1282)

Reviewer 2 Report

The authors have performed a descriptive cohort study on factors associated with cervical cancer survivors returning to work. The work uses extensive statistical analyses and concludes that RTW positively correlates with longterm survival in cervical cancer patients. However, I would recommend minor changes to the manuscript before publication. I appreciate the fact that despite being a purely statistical study, the authors have provided conclusions/discussions that are relevant to public health.

Comments - 

1) There are definite but minor grammatical errors. In some places, they distort the meaning of the sentence. The authors should rectify these errors on lines - 192, 194, 232, 

2) Formatting errors

Word breakage - this could be a result of the template provided by the journal but breaking a word "parti-cipation" across 2 lines is not recommended in manuscripts. 

What reference style is this? Typically it is
"I like to eat mango (ref)". and not "I like to eat mango. (ref)"

Format Tables 3 and 4 to be easier for reading. You can orient the page lengthwise and then provide tables. 

3) Line 49 - What is the prevalence of breast cancer vs cervical cancers in the population under study? What is the morbidity and mortality rate? 

4) What do the authors mean by "ref:" mentioned in the tables?

5) Line 190 - Sentence ambiguous. Did you mean "occupational status" as positive/negative? Needs further description. 

6) Line 197 - Did these non RTW cancers also have higher overall mortality rates?

7) Line 217 - I think the more technical term for "white" is Caucasian. Please use that term. Also, can you clarify which country was this study conducted in.

Author Response

Thank you for your positive comments on this manuscript. The responses to the points raised by the reviewers are listed as follows.

1) There are definite but minor grammatical errors. In some places, they distort the meaning of the sentence. The authors should rectify these errors on lines - 192, 194, 232

Response: Thank you for your valuable comments. We followed your recommendation and modified the last paragraph of the discussion section. (Page 9, line1160-1162, line1163-1175 and Page10, line1218-1219)

2) Formatting errors

Word breakage - this could be a result of the template provided by the journal but breaking a word "parti-cipation" across 2 lines is not recommended in manuscripts. 

What reference style is this? Typically it is
"I like to eat mango (ref)". and not "I like to eat mango. (ref)"

Response: Thank you for your valuable comments. We followed your recommendation and modified the last paragraph.

Format Tables 3 and 4 to be easier for reading. You can orient the page lengthwise and then provide tables. 

Response: Thank you for your critique. We followed your and another reviewer’s recommendation and modified Table 3 and Table 4 format, then moved the table to the appendix section in the last paragraph.

3) Line 49 - What is the prevalence of breast cancer vs cervical cancers in the population under study? What is the morbidity and mortality rate? 

Response: Thank you for the valuable comments. To our knowledge, all of those reference articles we cited in this paragraph didn’t mention the prevalence and overall mortality rates data of their study population. However, according to the Global Cancer Observatory (GCO) database of WHO in 2020, the worldwide prevalence of breast cancer is 47.5 per 100,000 and age-standardized mortality rates is 13.6 per 100,000. The worldwide prevalence of cervical cancers is 10.7 per 100,000 and age-standardized mortality rates is 7.3 per 100,000.

Database link: https://gco.iarc.fr/today/online-analysis-table?v=2020&mode=cancer&mode_population=continents&population=900&populations=900&key=asr&sex=2&cancer=39&type=1&statistic=1&prevalence=0&population_group=0&ages_group%5B%5D=0&ages_group%5B%5D=17&group_cancer=1&include_nmsc=1&include_nmsc_other=1#collapse-by_country

4) What do the authors mean by "ref:" mentioned in the tables?

Response: Thank you for the valuable comments. The “ref:” in the tables means the control group in the Cox proportional hazard regression model of each independent covariates. For example, “Monthly income (ref: <28800)” in the tables means we compared other monthly income groups (such as monthly income 28800-38200 and monthly income >38200) with monthly income <28800 group in our statistical model.

5) Line 190 - Sentence ambiguous. Did you mean "occupational status" as positive/negative? Needs further description. 

Response: Thank you for your careful review and constructive comments. The “occupational status” here means the condition of employment. However, in some of the articles, occupational status means the collective term encompassing occupational performance components, occupational performance, and occupational role performance. We followed your recommendation and modified the term to “employment status” in the last paragraph. (Page 9, line 1159)

6) Line 197 - Did these non RTW cancers also have higher overall mortality rates?

Response: Thanks for your thorough review and critique. According to this reference article “Returning to work after sick leave due to cancer: a 365-day cohort study of Japanese cancer survivors. J Cancer Surviv 2016, 10(2):320-329.”, it didn’t mention overall mortality rates of these non-RTW cancers in the study. However, according to the Global Cancer Observatory (GCO) database of WHO in 2020 overall mortality rates of different cancer in Japan are as follow:

Non-RTW cancer

age-standardized mortality rates

(per 100,000)

RTW cancer

age-standardized mortality rates

(per 100,000)

Lung cancer

14.7

Breast cancer

9.9

Esophageal cancer,

2.8

Gastric cancer

8.2

Hepatic cancer,

4.8

Ovary cancer

3.2

Pancreatic cancer

7.7

Cervical cancer

2.9

Non-Hodgkin lymphoma

2.6

Bladder cancer

1.6

Leukemia

2.4

Prostate cancer

4.5

Database link: https://gco.iarc.fr/today/online-analysis-table?v=2020&mode=cancer&mode_population=continents&population=900&populations=392&key=asr&sex=0&cancer=39&type=1&statistic=1&prevalence=0&population_group=0&ages_group%5B%5D=0&ages_group%5B%5D=17&group_cancer=1&include_nmsc=1&include_nmsc_other=1#collapse-group-0-3

7) Line 217 - I think the more technical term for "white" is Caucasian. Please use that term. Also, can you clarify which country was this study conducted in.

Response: Thank you for your insightful comments and helpful suggestions. First, we followed your recommendation and modified the last paragraph of the discussion section. (Page 9, line1185). Second, the study which you mention above was conducted in California, USA.

Reviewer 3 Report

This nationwide cohort study is interesting that returning to work after cervical cancer treatment had a beneficial effect on survival. This study suggests that interventions to improve the cervical cancer survivors’ ability to RTW is needed after cancer treatment

However, the following issues should be considered before consideration of publication:

  1. In abstract, aim. The purpose of this study was to study the relationship between RTW and Survival outcomes. In abstract, this part is not mentioned first, so it is understood as if different medical and occupational covariates are more important factors It would be better if the author matches the title and the aim of the abstract.

  1. Line 27. As mentioned above, the main aim in introduction seems to reveal the relationship between RTW and covariables, while the title is about return to work and survival rates. Please check again.

  1. Materials and Methods. 2.1 study population. Since insurance systems and the data collection methods of insurance systems differ from country to country, it may helpful to add a brief description of the Taiwan cancer registry or NHIRD. In addition, in order to estimate the sample size of this study, it would be helpful to mention statistical data of Taiwan such as the annual number of cervical cancer incidences in Taiwan.

  1. 4. Outcome measures. It is necessary to determine whether the primary outcome is RTW within the first 5 years or survival outcomes.

  1. 3.2 Characteristics of employment of cervical cancer survivors during the 5-year period. It would be good to mention the reason for comparing 2-year RTW and 5-year RTW. Also, it would be better if you add the reason why income is divided by 28800 and 38200.

  1. 3 Univariate association between RTW and independent variables. The control group should be clarified in statistics. For example, in the 3-year RTW group, it is not clear whether the control group is non-RTW or RTW group after 3 years.

  1. 5 Effect of RTW on survival outcomes in cervical cancer survivors. In Table5. Univariate analysis should be performed for several factors, and then multivariate analysis should be performed and the process should be added to the table.  

  1. The period of cancer treatment is an important factor. This is one of the limitations because the authors didn’t consider the treatment periods. If this limitation is mentioned in the discussion or additional analysis is performed with treatment period, this manuscripts will be a more meaningful study.

Author Response

Thank you for your positive comments on this manuscript. The responses to the points raised by the reviewers are listed as follows.

1. In abstract, aim. The purpose of this study was to study the relationship between RTW and Survival outcomes. In abstract, this part is not mentioned first, so it is understood as if different medical and occupational covariates are more important factors It would be better if the author matches the title and the aim of the abstract.

      Response: Thank you for your careful review and constructive comments. The aim of the current cohort study was to explore the relationship between return to work after cervical cancer treatment and different medical and occupational covariates. We also investigated the effect of RTW on all‐cause mortality and survival outcomes of cervical cancer survivors. We followed your recommendation and modified the title of the article to “The fact of the return to work in cervical cancer survivors and the impact on survival rate: An 11-year follow-up study.

2. Line 27. As mentioned above, the main aim in introduction seems to reveal the relationship between RTW and covariables, while the title is about return to work and survival rates. Please check again.

     Response: Thank you for your constructive comments. The aim of the current cohort study was to explore the relationship between return to work after cervical cancer treatment and different medical and occupational covariates. We also investigated the effect of RTW on all‐cause mortality and survival outcomes of cervical cancer survivors. We followed your recommendation and modified the title of the article to “The fact of the return to work in cervical cancer survivors and the impact on survival rate: An 11-year follow-up study.”

3. Materials and Methods. 2.1 study population. Since insurance systems and the data collection methods of insurance systems differ from country to country, it may helpful to add a brief description of the Taiwan cancer registry or NHIRD. In addition, in order to estimate the sample size of this study, it would be helpful to mention statistical data of Taiwan such as the annual number of cervical cancer incidences in Taiwan.

      Response: Thank you for your valuable comments. The Taiwan Cancer Registry (TCR) is a nationwide population-based cancer registry system that was established by the Ministry of Health and Welfare in 1979. The data of patients with newly diagnosed cancer in hospitals with 50 or more beds in Taiwan are collected and reported to the TCR. To evaluate cancer care patterns and treatment outcomes in Taiwan, the TCR established a long-form database in which cancer staging and detailed treatment and recurrence information have been recorded since 2002. Furthermore, in 2011, the long-form database began to include detailed information regarding cancer site-specific factors, such as laboratory values, tumor markers, and other clinical data related to patient care. According to the TCR data in 2018, the annual number of newly diagnosed cervical cancer in Taiwan is 1433 cases, and the incidence rate is 12.07 per 100,000 persons. We followed your recommendation and modified the material and methods section in the last paragraph. (Page2, line 76-77).

4. 4. Outcome measures. It is necessary to determine whether the primary outcome is RTW within the first 5 years or survival outcomes.

Response: Thank you for your valuable comments. The primary outcome for this study was RTW within the first five years after cervical cancer diagnosis.

5. 3.2 Characteristics of employment of cervical cancer survivors during the 5-year period. It would be good to mention the reason for comparing 2-year RTW and 5-year RTW. Also, it would be better if you add the reason why income is divided by 28800 and 38200.

      Response: Thank you for your insightful comments and helpful suggestions. The reasons for comparing 2-year RTW and 5-year RTW were as follow:

(1) Most of the cancer treatment will be complete within two years, so we considered a 2 year period that may represent the stable condition of the disease.

(2) According to the previous studies, cancer survivors usually returned to work within 2 years after their cancer diagnosis.

(3) Most of the studies use a five-year survival rate to represent cancer survival rates. 

Monthly income divided by 28800 and 38200 is due to the original Labor Insurance Database is provided in this form.  

6.3 Univariate association between RTW and independent variables. The control group should be clarified in statistics. For example, in the 3-year RTW group, it is not clear whether the control group is non-RTW or RTW group after 3 years.

      Response: Thank you for the valuable comments. The control group in the Cox proportional hazard regression model of each independent covariates is labeled as “ref:” in the tables. For example, “Monthly income (ref: <28800)” in the tables means we compared other monthly income groups (such as monthly income 28800-38200 and monthly income >38200) with monthly income <28800 group in our statistical model.

7. 5 Effect of RTW on survival outcomes in cervical cancer survivors. In Table5. Univariate analysis should be performed for several factors, and then multivariate analysis should be performed and the process should be added to the table.  

     Response: Thank you for your critique. We followed your recommendation and modified the table. (Page 8,9)

8. The period of cancer treatment is an important factor. This is one of the limitations because the authors didn’t consider the treatment periods. If this limitation is mentioned in the discussion or additional analysis is performed with treatment period, this manuscripts will be a more meaningful study.

     Response: Thank you for your careful review and constructive comments. We followed your recommendation and modified the discussion in the last paragraph. (Page10, line 1236-1237).